# Regulatory Effects and Mechanism of Action of Green Tea Polyphenols on Osteogenesis and Adipogenesis in Human Adipose Tissue-Derived Stem Cells

**Weiguo Lao, Yi Zhao, Yi Tan, Michael Johnson, Yan Li, Linda Xiao, Jing Cheng, Yiguang Lin *** and **Xianqin Qu ***

School of Life Sciences, University of Technology Sydney, Ultimo, NSW 2007, Australia
* Correspondence: yiguang.lin@hotmail.com (Y.L.); xianqin.qu@uts.edu.au (X.Q.)

**Abstract:** We previously showed that green tea polyphenols (GTPs) exert antiadipogenic effects on preadipocyte proliferation. Here, we investigated the regulatory effects of GTPs on osteogenesis and adipogenesis during early differentiation of human adipose tissue-derived stem cells (hADSC). Adipogenesis of hADSCs was determined by oil-red-O staining and triglycerides synthesis measurement. Osteoporosis of hADSC was measured using alkaline phosphatase assays and intracellular calcium levels. Immunofluorescence staining and qRT-PCR were used to detect PPARγ-CEBPA regulated adipogenic pathway regulated by PPAR-CEBPA and the osteogenic pathway mediated by RUNX2-BMP2. We found that GTPs treatment significantly decreased lipid accumulation and cellular triglyceride synthesis in mature adipocytes and attenuated pioglitazone-induced adipogenesis in a dose-dependent manner. GTPs downregulated protein and mRNA expression of *Pparγ* and attenuated pioglitazone-stimulated-*Cebpa* expression. GTPs treatment significantly enhanced hADSCs differentiation into osteoblasts compared to control and pioglitazone-treated cells. GTPs upregulated RunX2 and Bmp2 proteins and mRNA expression compared to control and significantly attenuated decreased RunX2 and Bmp2 mRNA expression by pioglitazone. In conclusion, our data demonstrates GTPs possesses great ability to facilitate osteogenesis and simultaneously inhibits hADSC differentiation into adipogenic lineage by upregulating the RUNX2-BMP2 mediated osteogenic pathway and suppressing PPARγ-induced signaling of adipogenesis. These findings highlight GTPs' potential to combat osteoporosis associated with obesity.

**Keywords:** human adipose tissue-derived stem cells; green tea polyphenols; adipogenesis; PPARγ-CEBPA signaling pathway; osteogenesis; RUNX2-BMP2 pathway



## 1. Introduction

Osteoporosis is a metabolic disease that affects bone mineral density and is one of the leading causes of bone fracture in the aging population with obesity [1,2]. The pathophysiology of osteoporosis is multifactorial. Accumulating data have shown that in obesity, the disease may be initiated by abnormal activation of adipogenesis in mesenchymal stem cells (MSC), which affects the differentiation of MSCs into osteoblasts, leading to the development of osteoporosis and metabolic disorders [3,4]. Thus, MSCs, including those derived from bone marrow (BMSC) and adipose tissue (AD-MSC), have received considerable attention to the understanding of the pathogenesis of osteoporosis in obesity.

Adipocytes and osteoblasts originate from a common ancestor–pluripotent mesenchymal stem cells and there is an inverse relationship between adipocytes and osteoblasts in the bone marrow [5]. Significantly, inducers of differentiation towards one lineage may inhibit cell differentiation into an alternative lineage. In the aging population with obesity, differentiation to adipocytes dominates over the differentiation to osteoblasts from BMSC and AD-MSC [6]. Therefore, the imbalance between osteogenesis and adipogenesis during stem cell differentiation contributes to osteoporosis and the tendency to bone fracture in

the elderly with obesity. Therefore, the molecular shift of stem cells towards osteogenesis and diminishing adipogenesis may prove to be an important therapeutic approach to osteoporosis [7].

There is growing evidence that nutritional intervention with phytochemicals is effective in controlling obesity and bone loss simultaneously [8]. Epidemiological studies revealed that post-menopausal habitual tea drinkers have higher bone mineral density (BMD) [7] and reduced risk of hip fractures among the elderly [9–11]. A number of studies showed that GTPs or epigallocatechin gallate (EGCG, one of the GTPs), inhibited AD-MSC differentiation into adipocytes and regulated energy metabolism and white/brown adipogenesis, lowering the risk of obesity [12,13]. Green tea catechins, the main polyphenols found in green tea, have also been reported to have osteoprotective roles [14], and EGCG increased osteogenic differentiation of murine bone marrow MSCs [15]. However, the precise mechanism of nutritive polyphenols in maintaining bone health is not fully defined. No studies have been conducted to assess how GTPs regulates osteogenesis and adipogenesis in stem cells derived from human adipose tissue (hADSCs).

Previously, we demonstrated that GTPs has antiadipogenic effects on preadipocyte proliferation in 3T3-L1 adipocytes [16] and ameliorates metabolic abnormalities and insulin resistance in obese Zucker-fatty rats [17]. In the present study, we extended our study to examine whether GTPs is capable of modulating the differentiation of hADSCs that shift towards the osteogenic lineage and suppress adipogenesis during the early stage of differentiation. Peroxisome proliferator-activated receptor gamma (PPARγ), a ligand-activated transcription factor, upregulates MSCs differentiation into adipocytes [18]. In this study, we included PPARγ agonist (pioglitazone) as a reference drug to clarify the molecular pathways by which GTPs regulates adipogenesis and osteogenesis during differentiation of hADSCs into mature adipocytes and osteoblasts. Biomarkers and genes involved in the adipogenic and osteogenic pathways were analyzed.

## 2. Materials and Methods

### 2.1. Green Tea Polyphenols and Pioglitazone

The preparation of GTPs (99% of total catechins consisting of: 70.9% epigallocatechin gallate (EGCG), 1.7% epigallocatechin, 7.4% epicatechin gallate and 19.3% epicatechin) was generously provided by Zuyi Lushen Kangyuan Co. (Zuyi, China). Pioglitazone (Pio) was purchased from Sigma-Aldrich (St. Louis, MO, USA) and was used to induce adipogenesis during hADSC differentiation.

### 2.2. Isolation and Culture of hADSCs

The hADSCs were generously provided by Dr. Jerran Santos from an existing research project approval under the Macquarie University Human Research Ethics Committee (Ref #: 5201100385). hADSCs were isolated with the method previously described [19]. After isolation, hADSCs were confirmed by negative CD45 and positive CD90 with fluorescein isothiocyanate kits (BD Biosciences, San Jose, CA, USA) using an FC500 flow cytometer (Beckman Coulter, Brea, CA, USA). Primary hADSCs were subcultured in Dulbecco modified Eagle medium (DMEM) containing 10% fetal bovine serum (FBS, Gibco Life Technologies, Auckland, New Zealand) with 1% penicillin-streptomycin (PS, Gibco Life Technologies, Auckland, New Zealand) for passages 5 to 6 for the following in vitro studies.

### 2.3. Differentiation and Treatment of hADSCs

To determine the optimal time for the differentiation of primary hADSCs into mature adipocytes, cells were seeded at a density of $5 \times 10^3$/well in a 24-well plate in growth medium (DMEM with 10% FBS and 1% PS). When the cells were expanded to 70% confluence (designated as day 0 of differentiation induction), growth medium containing Glutmax/F12 (Gibco Life Technologies, Auckland, New Zealand) with 0.5 mmol of isobutyl-methylxanthine (IBMX), 1 μmol dexamethasone, 10 μmol insulin, 200 μmol indomethacin was replaced for adipogenic induction. This adipogenic medium was refreshed every

2 days. The number of mature adipocytes was determined by visual inspection of the formation of lipid droplets under light microscopy (Olympus, BX51 microscope, Tokyo, Japan) followed by Oil Red O staining (Sigma-Aldrich, St Louis, MO, USA) and measurement of the triglycerides described previously [16] on days 7, 14, 21 and 28 after induction treatment. Triglyceride content increased more than 10 times on day 21 of adipogenic induction.

For the determination of the optimal time for osteogenic differentiation, growth medium containing 0.1 μmol dexamethasone, 50 μmol ascorbate-2-phosphate, 10 mmol β-glycerophosphate was added to the cell culture and replaced every 2 days. The mature osteocytes were confirmed by measuring the activity of alkaline phosphatase (ALP) and calcium content at each point in the experiment the same as adipogenic induction. Calcium levels reached almost 10-fold higher than the baseline level in cells on day 14 after osteogenic induction.

According to above time course study, GTPs treatment for adiogenesis and osteogenesis was scheduled for 21 days and 14 days after induction, respectively. Briefly, GTPs (99% purity) was first dissolved at 10 mg/mL in sterile distilled water with 0.1% dimethylsulfoxide (DMSO) and this preparation was used to achieve the final concentrations at 1 and 10 μg/mL in hADSC culture in the presence or absence of 100 μmol of pioglitazone. Fresh induction medium with treatment reagents was replaced every 2 days.

### 2.4. Determination of Lipid Accumulation and Triglycerides Content

Oil Red O (Sigma-Aldrich, St Louis, MO, USA) staining was used to detect lipid droplets in differentiated adipocytes. Cells were washed with phosphate-buffered saline (PBS) three times and fixed with 10% formalin at room temperature for 1 h. After fixation, cells were washed once with PBS and stained with filtered Oil Red O solution (60% isopropanol, 40% water) for 30 min. After the lipid droplets were staining, the Oil Red O staining solution was removed, and the plates were rinsed with water to remove unbound dye, dried, and photographed. The stained lipid droplets were viewed with an Olympus microscope (Tokyo, Japan) and the images were captured with a digital camera (DP70, Tokyo, Japan), then quantitated using Image-Pro6.2 software (Media Cybernetics, Inc. Rockville, MD, USA) from 4–5 individual experiments.

Quantification of cellular triglyceride content was carried out on day 21 after differentiation induction and treatment. After washing with PBS, differentiated cells were harvested with 200 μL PBS and then centrifuged at $1000 \times g$ for 10 min. The pellet was dissolved in 1 mL of lipid extracting solution (chloroform:methanol = 2:1). After drying under nitrogen gas, the lipids were lysed in 100 μL ethanol. The concentration of triglycerides in the cell lysates was quantified using the commercial triglyceride assay kit (Wako Pure Chemical Industries, Osaka, Japan) of 5 individual experiments. The concentration of triglycerides was calculated according to the standard curve. The cellular content of triglycerides was adjusted based on the quantity of protein.

### 2.5. Alkaline Phosphatase Assay and Determination of Intracellular Calcium

Alkaline phosphatase (ALP) is expressed in the early stages of differentiation and is a marker of the osteoblastic phenotype. hADSCs were cultured in osteogenic differentiation medium with a variety of treatments for 14 days. At the end of treatment, cells were harvested from each well with 0.05% trypsin-0.53 mmol of EDTA and washed with PBS, lysed in 1 mL of 0.2% Triton X-100 aqueous solution using a Vibra-Cell sonicator (VXC 500 series, Sonic and Materials Inc., Newtown, CT, USA) for 30 s. The sonicates were centrifuged at $10,000 \times g$ for 10 min and the supernatants were collected for the ALP activity assay using a Chemistry Analyzer (Abbott Architect ci16200, Abbott Park, IL, USA) according to the manufacturer's instructions.

To detect intracellular calcium content, harvested cells were dissolved in 200 μL of 0.5 M HCl and then vigorously shaken for 16 h at room temperature, then the mixtures were sonicated for 30 s followed by centrifugation at $5000 \times g$ for 10 min. The supernatant was collected for the calcium assay using a Chemistry Analyzer (Abbott Architect ci16200,

Abbott Park, IL, USA) followed the manual instructions. The calcium content was calibrated with the number of cells counted in each well.

### 2.6. Immunofluorescence Staining and Quantification

PPARγ protein expression in adipogenic differentiation and runt-related transcription factor 2 (Runx2) protein expression in osteogenic differentiation were determined by immunofluorescence staining. Primary hADSCs were grown cells ($2 \times 10^4$ cells/well) were seeded on glass coverslips in 24-well plates until 70% confluence then maintained for 14 days of osteogenic induction and 21 days of adipogenic induction with a variety of treatments. At the end of treatment, cells were washed with cold PBS and then fixed in 4% paraformaldehyde for 20 min, permeabilized with PBS-T solution (PBS with 0.06% Tween 20 and 0.04% Triton 100) for 5 min, blocked with 5% bovine serum albumin (BSA) for 30 min. After blocking, cells were incubated with primary antibody overnight at 4 °C. Primary antibodies and dilutions were: rabbit anti-PPARγ (Santa Cruz Biotechnology, Dallas. TX, USA 1:100) or mouse anti-Runx2 monoclonal (Cell Signaling Technology, Danvers, MA, USA 1:100) in 2% BSA-PBS and incubated overnight at 4 °C. A 2% BSA-PBS solution was used for negative control. After incubation with the primary antibody, cells were washed with fresh PBS followed by staining with secondary antibodies: goat anti-rabbit or goat anti-mouse IgG Alexa Fluor 488 (Thermo Fisher Scientific, Waltham, MA, USA 1:400) for 1 h at room temperature in the dark. After washing with PBS for 3 times, the coverlip containing the cells was mounted on glass slides with SlowFade Diamond Antifade Mounting with Propidium Iodide (PI) (Gibco Life Technologies, Auckland, New Zealand) for 2 min at room temperature. Images were obtained using a fluorescent microscope (Olympus BX51 and DP70, Tokyo, Japan) and the fluorescent intensity of the antibody-labelled regions within cells was calculated from the images using Image J software (National Institutes of Health, Bethesda, MD, USA) from 4 individual experiments.

### 2.7. Total RNA Isolation and Quantitative Real-Time PCR

To further clarify the molecular mechanism by which GTPs modulates the differentiation of hADSC, the gene expression of *Pparγ*, enhancer-binding protein alpha *(Cebpa)* and the cyclic adenosine monophosphate (cAMP) responsive element binding protein *(Creb) that* regulate adipogenesis, as well as the *Runx2* and bone morphogenetic protein 2 (*Bmp2*) genes that regulate osteogenesis, was determined by quantitative real-time polymerase chain reaction (qRT-PCR).

Amplification of complementary DNA (cDNA) with primers (Thermo Fisher Scientific, Waltham, MA, USA) shown in Table 1. cDNA was carried out using Stratagene MXPro-Mx3000P (Agilent Technologies, Waldbronn, Germany). qRT-PCR was performed with the QuantStudio 6k Flex Real-Time System (Applied Biosystems, Foster City, CA, USA) using SYBR FastStart Universal SYBR Green Master Mixes (Roche, Mannheim, Germany) according to the manufacturer's instructions. The Delta Ct value (Ct [targeted gene]-Ct [Actb]) was used as a measure of relative changes messenger RNA (mRNA) of *Cebpa*, *Pparγ*, *Creb*, *Runx2* and *Bmp2* in GTPs treated cells and compared to the mRNA levels of positive and negative controls. Each sample was amplified in triplicate and the expression of beta-actin (*Actb*) was used as an internal control for every PCR assay.

**Table 1.** Primer sequences used in real-time PCR.

| Primer Symbol | Forward Primer | Reverse Primer |
|:---:|:---:|:---:|
| *Cebpa* | 5′-TATAGGCTGGGCTTCCCCTT-3′ | 5′-AGCTTTCTGGTGTGACTCGG-3′ |
| *Pparγ* | 5′-CCGTGGCCGCAGATTTGA-3′ | 5′-AGATCCACGGAGCTGATCCC-3′ |
| *Creb* | 5′-TTCAAGCCCAGCCACAGATT-3′ | 5′-AGTTGAAATCTGAACTGTTTGGAC-3′ |
| Runx2 | 5′-CACCGAGACCAACAGAGTCA-3′ | 5′-TGGTGTCACTGTGCTGAAGA-3′ |
| Bmp2 | 5′-TTTCAATGGACGTGTCCCCG-3′ | 5′-AGCAGCAACGCTAGAAGACA-3′ |
| Actb | 5′-CTCACCATGGATGATGATATCGC-3′ | 5′-AGGAATCCTTCTGACCCATGC-3′ |

*2.8. Statistical Analyses*

Statistical analyses were performed using GraphPad Prism version8.0 (GraphPad Software, San Diego, CA, USA) and all values are expressed as means ± SE. Differences between groups were examined using one-way analysis of variance (ANOVA) followed by Tukey's test to determine significant differences between groups. The *p* value < 0.05 were considered statistically significant.

**3. Results**

*3.1. GTPs Reduces Lipid Accumulation and Triglyceride Synthesis during hADSC Differentiation into Adipocytes*

To examine the effects of GTPs on adipocyte differentiation, lipid accumulation was determined on day 21 of differentiation cells with 1 and 10 μg/mL of GTPs alone or in combination with 100 μmol of pioglitazone. Figure 1A showed the intracellular lipid droplets stained with Oil Red O and the quantitative results of the accumulation of lipids (Figure 1B). Pioglitazone significantly increased ($p < 0.001$) the total amount of lipids compared to control cells. GTPs prevented lipid accumulation (both $p < 0.05$ vs. control cells). Furthermore, GTPs attenuated Pio-induced lipid accumulation in a dose-dependent manner ($p < 0.05$ and $p < 0.01$ vs. pioglitazone alone) during ADSC differentiation to adipocytes.

In agreement with the reduction of total lipid content, intracellular triglyceride levels were reduced ($p < 0.05$) in GTPs treated cells compared to control cells. Pioglitazone stimulated triglyceride production ($p < 0.01$ vs. control) and this was markedly attenuated by the addition of GTPs (Figure 1C).

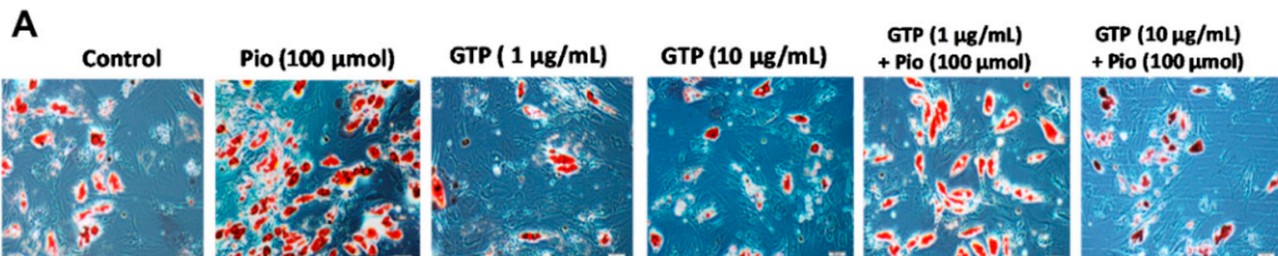

**Figure 1.** *Cont.*

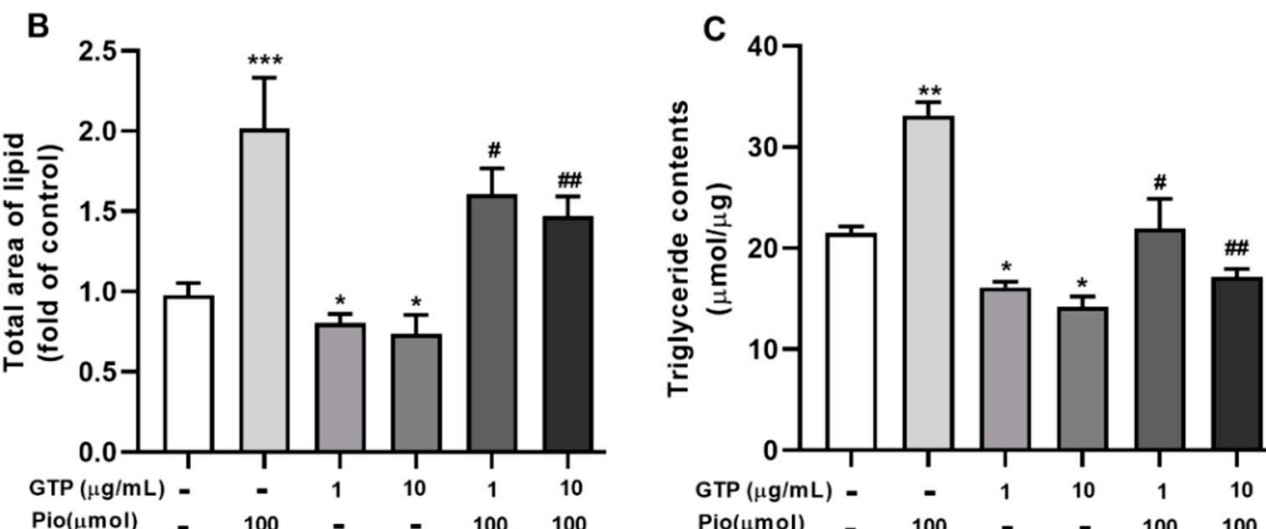

**Figure 1.** Green tea polyphenols (GTPs) decrease lipid accumulation and triglyceride synthesis during adipogenic differentiation of hADSC. (**A**). Image of the lipid droplets with Oil Red O (ORO) staining and examined by light microscopy at 100×, objective magnification; (**B**). Quantitative data of the total area occupied by lipid droplets stained with ORO; (**C**). Triglyceride content was determined by measuring absorbance at 490 nm. Data are presented as means $\pm$ SE ($n$ = 5 per group). * $p < 0.05$, ** $p < 0.01$ and *** $p < 0.001$ GTPs and pioglitazone vs. the control; # $p < 0.05$ and ## $p < 0.01$ GTPs plus pioglitazone (Pio) vs. Pio alone.

### 3.2. GTPs Inhibits the Expression of PPARγ Protein and Genes Involved in ADSCs toward Adipogenic Differentiation

PPARγ is a key transcription factor that up-regulates the expression of genes involved in adipogenesis. To elucidate the mechanism by which GTPs inhibits adipogenesis during ADSC differentiation, PPARγ expression was detected with immunofluorescence staining and PPARγ agonist (pioglitazone) was used as a reference drug. Figure 2 shows that pioglitazone at 100 µmol stimulated the expression of PPARγ by 24.1 $\pm$ 0.78% compared to the control ($p < 0.05$). Treatment with GTPs treatment at 1 and 10 µg/mL downregulated PPARγ expression by 22.1 $\pm$ 0.97% ($p < 0.05$) and 44.3 $\pm$ 1.3% ($p < 0.01$), respectively, compared to control. When GTPss were added to pioglitazone, increased PPARγ expression of PPAR decreased by 24.3 $\pm$ 1.03% with 10 µg/mL of GTPs ($p < 0.05$), suggesting that GTPs prevented pioglitazone-induced PPARγ overexpression.

PPARγ, CCAAT-enhancer-binding proteins (C/EBPα) and CREB play a crucial role in the adipogenic pathway. To define the molecular mechanism by which GTPss negatively regulate hADSCs toward adipocyte differentiation, transcription factors of *Pparγ*, *Cebpa* and *Creb* were analyed with qRT-PCR. Pioglitazone significantly increased the expression of *Pparγ* ($p < 0.01$) and *Cebpa* ($p < 0.05$) but did not significantly affect the mRNA of *Creb*. Low-dose GTPs at low dose (1 µg/mL) did not affect the expression of these transcription factors, but 10 µg/mL of GTPs negatively regulated the expression of Ppar ($p < 0.05$) when compared to the control (Figure 3A). Treatment with GTPs treatment prevented the expression of pioglitazone-stimulated *Cebpa mRNA* (Figure 3B, $p < 0.05$) and *Creb mRNA* (Figure 3C, $p < 0.05$).

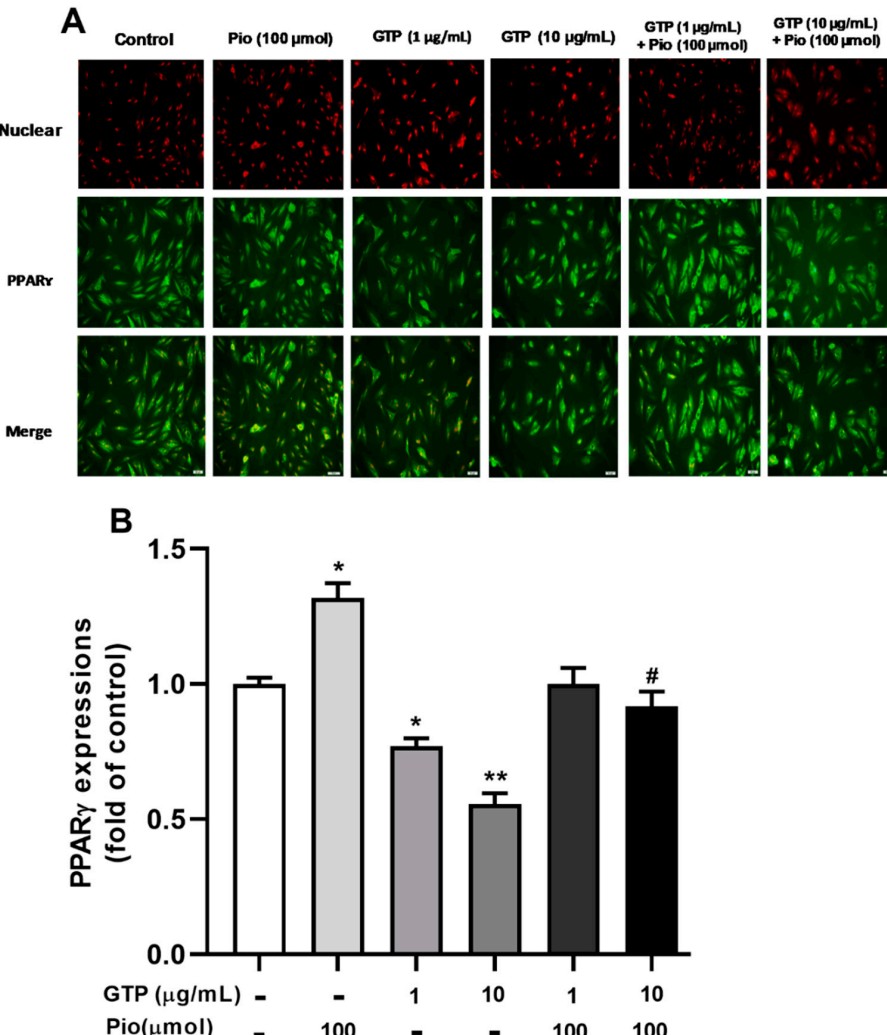

**Figure 2.** Green tea polyphenols (GTPs) inhibit PPARγ expression during adipogenic differentiation of hADSCs. Immunofluorescence staining images (**A**) depicted a marked elevation in the uorescent intensity of PPARγ antibodies conjugated to FITC stained with green color in cells treated with pioglitazone (Pio, 100 μmol) treatment and a reduction in the fluorescent intensity of PPAR antibodies conjugated to FITC in cells treated with GTPs (1 and 10 μg/mL). Quantitative data (**B**) are expressed as means ± SE. * $p < 0.05$ and ** $p < 0.01$ GTPs and pioglitazone treatments vs. control. # $p < 0.05$ GTPs plus pioglitazone vs. pioglitazone alone.

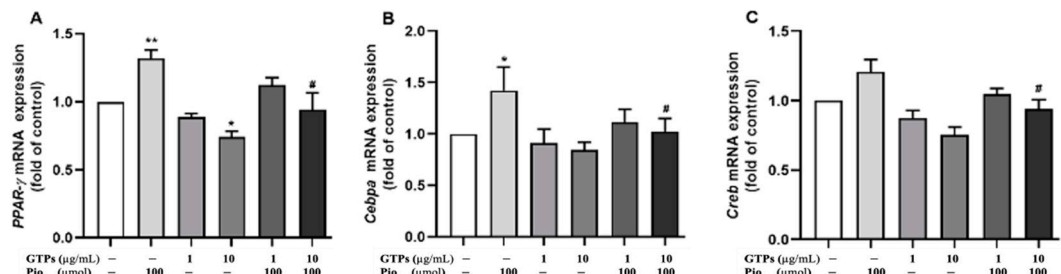

**Figure 3.** Effects of green tea polyphenols (GTPs) on mRNA expression of *Pparγ Cebpa* and *Creb*. Gene expression of *Pparγ* (**A**), *Cebpa* (**B**) and *Creb* (**C**) was detected by qRT-PCR in cell cultures with DMEM (control), pioglitazone (Pio, 100 μmol), GTPs (1 and 10 μg/mL) and GTPs plus Pio for 21 days. Data from 4 separate experiments are expressed as means ± SE. * $p < 0.05$ and ** $p < 0.01$ GTPs and pioglitazone treatments vs. control. # $p < 0.05$ GTPs plus pioglitazone vs. pioglitazone alone.

### 3.3. GTPs Stimulates Mineralization and ALP Activity during Osteogenic Differentiation of hADSCs

Calcium deposition and ALP activity were used to characterize the mineralization and osteogenic differentiation of hADSCs toward mature osteoblasts. Figure 4 shows the calcium content and ALP activity in cells on days 14 after osteogenic induction with a variety of treatments. Pioglitazone (Pio, 100 μmol) significantly decreased calcium content (Figure 4A, $p < 0.05$ vs. the control) and inhibited ALP activity by $25.0 \pm 3.4\%$ (Figure 4B, $p < 0.05$ vs. the control). GTPs at 10 μg/mL significantly improved calcium content (Figure 4A, $p < 0.05$ vs. control) associated with increased APL activity (Figure 4B, $p < 0.05$) but GTPs at 1 μg/mL neither significantly influence intracellular calcium levels or changed ALP activity in mature osteogenic cells, suggesting a higher dose of GTPs is required to achieve osteogenic effect. Furthermore, GTPs treatment at 10 μg/mL partially reversed pioglitazone-reduced calcium deposition (Figure 4B, $p < 0.05$ vs. Pioglitazone alone) and ALP activity (Figure 4B, $p < 0.05$ and $p < 0.01$, respectively, vs. pioglitazone alone).

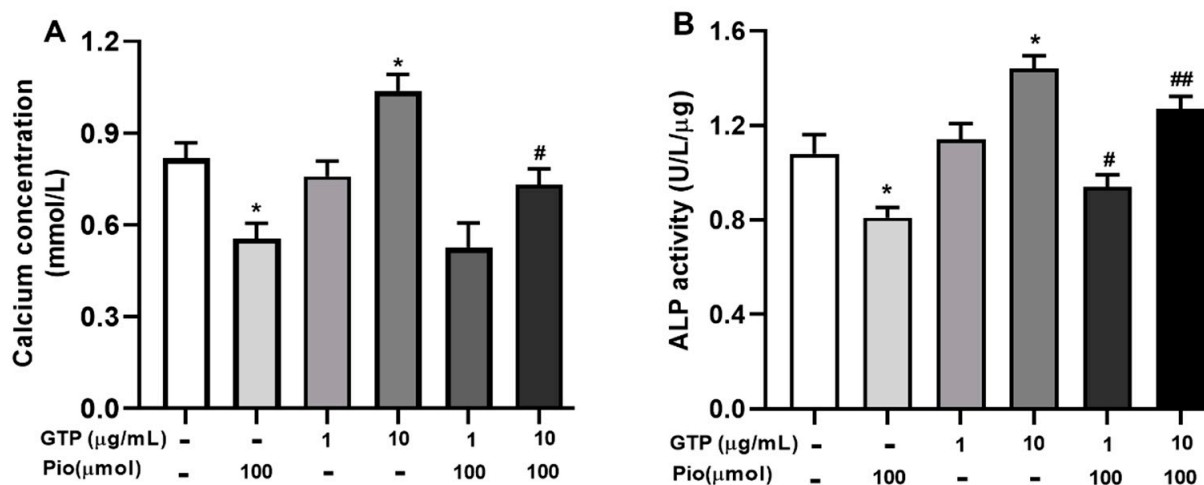

**Figure 4.** Green tea polyphenols (GTPs) increase calcium content and ALP activity in mature osteocytes. Quantitative assay of calcium level (**A**) and ALP activity (**B**) was performed using an automatic biochemistry instrument (ARCHITECT, Japan). Data from 4 separate experiments are presented as means $\pm$ SE. * $p < 0.05$ pioglitazone and GTPs vs. the control; # $p < 0.05$ and ## $p < 0.01$ GTPs plus pioglitazone vs. pioglitazone treatment alone.

### 3.4. GTPs Induces the Expression of Runx2 Protein and Genes Involved in ADSC towards Osteogenic Differentiation

Runx2 is a key protein molecule involved in the osteogenic differentiation process. To elucidate the mechanism by which GTPs enhances osteogenic differentiation, as evidenced by increased calcium content and ALP activity, Runx2 protein expression was measured with immunofluorescence staining on day 14 after differentiation induction and treatments. Figure 5 shows that pioglitazone slightly decreased Runx2 protein expression, and GTPs treatments (1 and 10 μg/mL) significantly enhanced Runx2 expression by $31.1 \pm 1.4\%$ ($p < 0.01$) and $37.2 \pm 2.1\%$ ($p < 0.001$) compared to control. Adding GTPs to pioglitazone slightly increased Runx2 expression compared to pioglitazone treatment alone; however, the enhancement did not achieve statistical significance even at a high dose of GTPs (10 μg/mL).

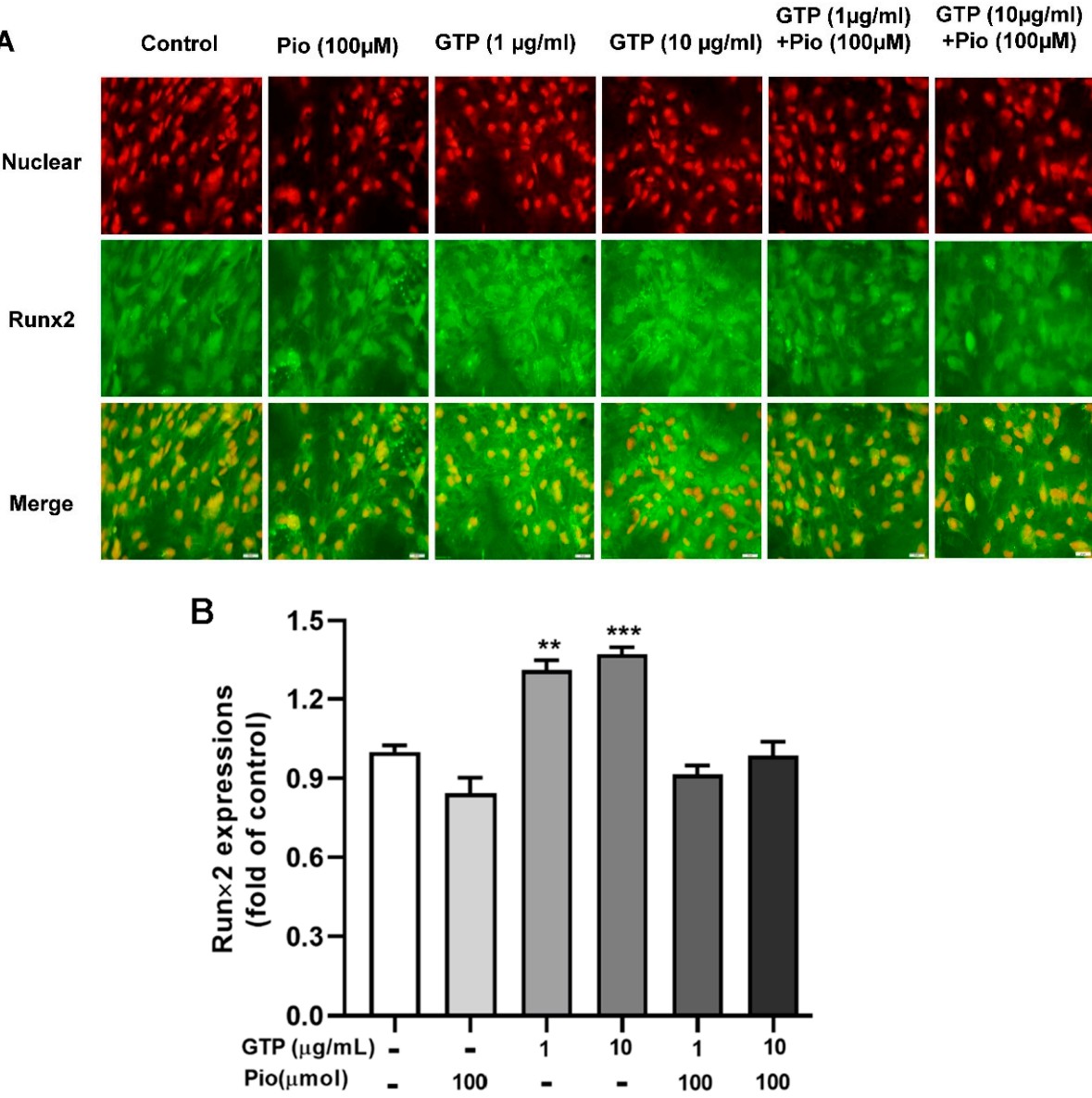

**Figure 5.** Green tea polyphenols (GTPs) enhance Runx2 protein expression during hADSC toward oestrogenic differentiation. Images of immunofluorescence staining (**A**) showed Runx2 protein contents stained as green in cells treated with pioglitazone (Pio, 100 μmol), 1 and 10 μg/mL of GTPs and the combination of GTPs and pioglitazone. Quantitative data (**B**) from 4 separate experiments are expressed as means ± SE. ** $p < 0.01$ and *** $p < 0.001$ GTPs treatments vs. the control.

To understand the molecular mechanism of GTPs-enhanced osteogenic differentiation from hADSCs, qRT-PCR detected *Runx2* and *Bmp2* mRNA expression. Treatment with pioglitazone did not significantly affect the expression of *Runx*2 and *Bmp*2 (Figure 6A,B). Interestingly, treatment with 10 μg/mL of GTPs significantly increased the expression of *Runx2* ( $p < 0.05$ ) and *Bmp*2 ( $p < 0.01$ ) compared to the control (Figure 6). The high dose of GTPs (10 μg/mL) added to pioglitazone also significantly increased *Runx2 expression* ( $p < 0.05$ ) and *Bmp2 expression* ( $p < 0.05$ ) expression compared to pioglitazone alone.

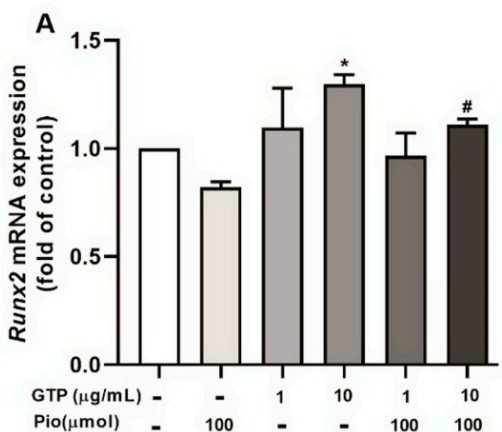
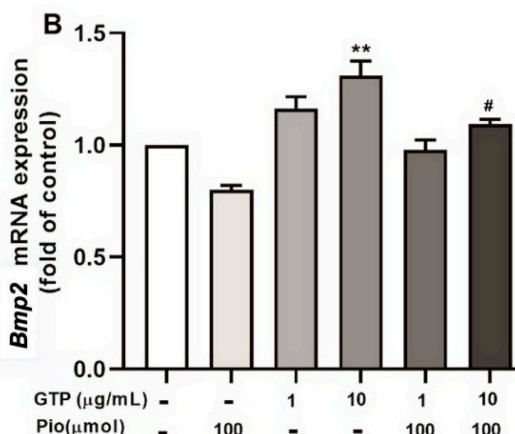

**Figure 6.** Green tea polyphenols (GTPs) upregulate Runx2 and Bmp2 mRNA expression in mature osteocytes. The gene expression of Runx2 (**A**) and Bmp2 (**B**) analyzed with qRT-PCR in the cell underwent 14 days of oestrogenic induction with a variety of treatments. Data from 4 separate experiments are expressed as means $\pm$ SE. * $p < 0.05$ and ** $p < 0.01$ GTPs treatments vs. control. # $p < 0.05$ GTPs plus pioglitazone vs. pioglitazone alone.

## 4. Discussion

The growing evidence supports a complex relationship between adiposity and osteoporosis in overweight/obese individuals, with an increase in adipogenesis accompanied by a decrease in bone formation in obese populations [19,20]. Osteoblasts and adipocytes differentiate from a common precursor, pluripotent mesenchymal stem cells found in bone marrow and adipose tissue, and have a reciprocal relationship between adipogenesis and osteogenesis at the early stage of differentiation [21]. Therefore, the tightly controlled lineage commitment of MSCs plays a critical role in maintaining bone homeostasis and lean body composition. Down-regulation of differentiation toward the adipogenesis lineage may promote cell differentiation into the osteogenesis lineage.

In the present study, we evaluated whether GTPs is capable of regulating the differentiation of primary human adipose-derived stem cells shifting towards osteogenic lineage and preventing adipogenesis. The results showed that GTPs prevents lipid accumulation and reduces triglyceride content in ADSCs destined to differentiate in adipocytic cells following exposure to an adipogenic differentiation medium. This finding is consistent with previous reports that EGCG, a major component of GTPs, has the ability to suppress adipogenesis through downregulating gene expression in mice MSCs [22] and to inhibit the differentiation of ADSC into adipocytes via altering STAT3 phosphorylation status [13]. Human ADSCs used in this study possess a higher stem cell population compared to other MSC sources [21] and its pluripotency behavior and pathophysiology are more correlative to human obesity than MSCs from mice [22]. This study demonstrated that GTPs exert an inhibitory effect on hADSCs adipocyte-differentiation. Furthermore, GTPs attenuated pioglitazone-stimulated adipogenesis. It is well known that pioglitazone as a PPARγ agonist can enhance insulin sensitivity for type 2 diabetes but adversely affects body composition and bone metabolism [18,23]. Our previous study has shown that GPT ameliorates metabolic abnormalities and insulin resistance by enhancing insulin signaling in skeletal muscle of Zucker fatty rats [17]. Taken together, the addition of GTPs to pioglitazone can exert a synergetic effect on insulin resistance, hyperglycemia, and metabolic complications of obesity.

An important finding of our study was that GTPs facilitated the differentiation of hADSCs into osteoblasts, a positive osteoprotective role of GTPs, in addition to its antiadipogenic effect. Firstly, GTPs treatment enhanced cellular calcium content, which is an essential substance for mineralization in mature osteoblasts. Secondly, ALP activity was higher in ADSC culture after 14 days of exposure to osteogenic differentiation medium

with GTPs treatment. ALP is a specific osteogenic marker in the early stages of osteogenic differentiation of MSCs, and its activity plays an important role in the mineralization of osteoblasts. In this study, the PPARγ agonist pioglitazone negatively regulated ADSC differentiation into osteoblasts, evidenced by reduction of calcium content and ALP activity in hADSC culture with 100 μmol of pioglitazone. Interestingly, GTPs treatment normalized pioglitazone-reduced calcium and ALP activity. An in vivo study by Shen et al. demonstrated that green tea supplementation prevented body weight gain and improved bone microstructure and strength in obese female rats fed a high-fat diet [24,25]. The results of the present study together with the findings of animal studies [17,24,25] indicate that the consumption of GTPs can simultaneously prevent fat formation and enhance bone formation. Furthermore, the combination of GTPs with PPARγ agonist may be an optimal therapy for the diabetic patient associated with obesity and osteoporosis.

To gain insight into the mechanism of GTPs promoting osteogenesis and inhibiting adipogensis, we examined the expression of molecular protein and genes that regulate the early stage of differentiation of hADSCs. PPARγ and the CCAAT enhancer binding protein family (C/EBPα, C/EBPβ, C/EBPδ), particularly C/EBPα, and the cAMP regulatory element binding protein (CREB) have been identified as master regulators involved in transcriptional control of the early stages of differentiation of adipocytes [26]. Activated PPARγ induces the expression of C/EBPα, which activates several genes involved in ADSC differentiation to mature adipocyte [27]. Immunofluorescence staining and qRT-PCR revealed that pioglitazone stimulates the expression of PPARγ and C/*ebpα*. Treatment with GTPs treatment at 10 μg/mL significantly reduced the expression of PPARγ protein and mRNA, indicating an inhibition of *Pparγ* gene and/or a reduction of transcription from mRNA to PPARγ protein with GTPs. GTPs did not affect the expression of *Cebpa* and *Creb*, but GTPs prevented pioglitazone-induced *Cebpα* overexpression. These results critically revealed a mechanism by which GTPs inhibits the differentiation of hADSCs into adipocytes by suppressing PPARγ–CEBPα mediated by PPAR-CEBP.

Recent studies have shown that there is an inverse relationship in adipogenic and osteogenic lineage commitment and differentiation [28]. In this study, we speculated that GTPs switching PPARγ-mediated adipogenesis towards osteoblast differentiation could induce biomarkers in controlling osteogenic differentiation, such as Runx2 and bone morphogenetic protein (BMP). The result of this study showed that pioglitazone negatively regulates osteogenesis by down-regulating Runx2 expression. Runx2 is a key gene expressed in the early stages of the osteogenic differentiation process [29]. Increased expression of Runx2 and mRNA in cells treated with GTPs indicates its osteogenic activity. Treatment with GTPs treatment also increased *Bmp2* expression. *Runx2* and *Bmp2* are critical molecular switches involved in osteogenic differentiation of mesenchymal stem cells [30,31]. These results indicate that GTPs is capable of promoting osteogenic differentiation by upregulating Runx2 and Bmp2 gene expression and inhibiting adipogenic differentiation of ADSCs by downregulating the expression of *Pparγ*.

The limitations of our study included that only limited steps of signaling pathways were investigated in the study. More comprehensive studies of the signaling pathway are needed, including studies using specific inhibitors to validate the involvement of specific cellular and molecular pathways. Secondly, our study only used osteoblast markers to study the differentiation of hADSCs into osteoblasts, the steps further to differentiate into osteoblasts have not been included in this study. Thirdly, our study used GTPs, rather than a specific component of GTPs. Therefore, the action of GTPs is not specifically related to a specific compound. Finally, the data were obtained from in vitro experiments only. More in vivo studies are certainly needed to move to the translational stage. Although promising, further in vivo studies, translational work and pharmacokinetic and toxicological assessments are necessary before moving on to clinical trials. We believe that the limitations are outweighed by the notable strengths and the promising future outlook of the potential new treatment strategy.

## 5. Conclusions

In summary, our data supports that during early differentiation of hADSC, GTPs plays a distinct regulatory role in the osteogenesis and adipogenesis, simultaneously facilitating osteogenesis and inhibiting differentiation into the adipogenic lineage. The possible underlying mechanism is through upregulating the RUNX2-BMP2 mediated osteogenic pathway and suppressing PPARγ-induced signaling of adipogenesis. The findings of this study highlight that GTPs may be a unique and promising therapeutic intervention for individuals with degenerative bone disorders or obesity and conditions that combine osteoporosis with obesity. The importance of adequate GTPs intake in combating chronic diseases such as osteoporosis and obesity is also implicated.

**Author Contributions:** Conceptualization, X.Q. and Y.L. (Yiguang Lin); methodology, W.L., Y.Z., Y.T., M.J., Y.L. (Yan Li), L.X. and J.C., software, X.Q., M.J. and Y.L. (Yiguang Lin); formal analysis, W.L., Y.Z., Y.T., M.J., Y.L. (Yan Li), L.X., J.C., Y.L. (Yiguang Lin) and X.Q.; investigation, W.L., Y.Z., Y.T., Y.L. (Yan Li), M.J., L.X., J.C. and Y.L. (Yiguang Lin); resources, X.Q., M.J. and Y.L. (Yiguang Lin); data curation, W.L., Y.Z., Y.T., M.J., Y.L. (Yan Li), L.X. and J.C.; writing—original draft preparation, X.Q. and Y.L. (Yiguang Lin); writing—review and editing, X.Q., Y.L. (Yiguang Lin) and M.J.; visualization, W.L., X.Q., M.J. and Y.L. (Yiguang Lin); supervision, Y.L. (Yiguang Lin) and X.Q.; project administration, X.Q.; funding acquisition, X.Q. All authors have read and agreed to the published version of the manuscript.

**Funding:** This research was funded by the special international collaboration grant (S2011GR0387) from the Ministry of Science and Technology of the People's Republic of China.

**Institutional Review Board Statement:** Not applicable.

**Informed Consent Statement:** Not applicable.

**Data Availability Statement:** The data (figures and tables) used to support the findings of this study are included within the article.

**Acknowledgments:** The authors are grateful to Zuyi Lushen Kangyuan Co. (Meitan, China) for supporting this study by providing green tea polyphenols. The authors would also like to thank Yali Sun's advice for chemical analysis of green tea polyphenols.

**Conflicts of Interest:** The authors declare no conflict of interest.

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
