# Peer review of "Regulatory Effects and Mechanism of Action of Green Tea Polyphenols on Osteogenesis and Adipogenesis in Human Adipose Tissue-Derived Stem Cells"

_cimb, doi:10.3390/cimb44120412_

Round 1
Reviewer 1 Report
Report
In this study, Weiguo Lao and the coauthors investigated regulatory effects of green tea polyphenols (GTP) on osteogeneic and adipogenic differentiation of human adipose tissue-derived stem cells (hADSCs).
They showed that GTP promotes osteogenic differentiation of hADSCs by inhibiting adipogenic differentiation. They also showed that the effects were exerted by upregulating the RUNX2-BMP2 mediated osteogenic pathway and suppressing PPAR. Although the results are clear and convincing, the following papers already showed the involvement of epigallocatechin-3-gallate (EGCG), which is a major polyphenol extracted from green tea, on both osteoblastogenic and adipogenic differentiation of hADSCs: (Zhang J, Wu K, Xu T, Wu J, Li P, Wang H, Wu H, Wu G. Epigallocatechin-3-gallate enhances the osteoblastogenic differentiation of human adipose-derived stem cells. Drug Des Devel Ther. 2019 Apr 23;13:1311-1321. PMID: 31114166.)
Therefore, the reviewer considered that this paper would not reach a high enough priority to be accepted for publication in this journal.
Author Response
Thank you for your comments.

Reviewer 2 Report
The study reported that green tea polyphenols can prevent adipogenesis and enhance osteogenesis in human adipose tissue-derived stem cells. The writing of the manuscript is clear and the data are well presented. However, I have several queries which need authors' clarification:
1. the authors mentioned at multiple points in the text that hADSC in osteogenic medium differentiated into osteocytes (instead of osteoblasts). Yet, no osteocyte markers are being tested. Hence, I think it is inappropriate to mention that they develop into osteocytes.
2. I wonder why intracellular calcium was measured, rather than calcium deposition on the culture plate. That would better reflect the mineralisation activity of the cells.
3. I think the authors should mention the limitations of the current study. The mechanistic studies are very preliminary and no inhibitors are used to validate the involvement of certain cellular signalling pathways.
4. please provide the lot number of antibodies used in the study.
Author Response
Thank you for your constructive comments.
We are really appreciate them.

Reviewer 3 Report
The ms explores the green tea polyphenols on osteogenesis and adipogenesis in humans adipose tissue-derived stem cells. The ms is well-written and correctly supported by data. However, I wonder if the authors can suggest some results utilizing single purified polyphenol. Moreover, conclusion could be better focused on obtained results, suggesting applications for GTP.
Round 2
Reviewer 1 Report
Thank you for your comments. However, this paper seems to be an adaptation of work by Zhang et al. Then the reviewer considered that this paper would not reach a high enough priority to be accepted for publication in this journal.
Author Response
Our response can be found in the attached PDF file.

Round 3
Reviewer 1 Report
Thank you for your comments. However, the reviewer considered that this paper would not reach a high enough priority to be accepted for publication in this journal.
Author Response
It is very hard for us to respond further as the reviewer does not have any further concerns and keeps repeating the same recommendation using the same sentences without justification. We believe that we have provided detailed responses and solid scientific evidence to clear up all the concerns that the reviewer had. A fair reviewer would always justify the recommendation made by providing a convincing reason/ground why such a recommendation is made. It is not clear why this reviewer repeats the same recommendation after reading our previous two detailed responses.